# Swimming of Spermatozoa in a Maxwell Fluid

**DOI:** 10.3390/mi10020078

**Published:** 2019-01-24

**Authors:** Toshihiro Omori, Takuji Ishikawa

**Affiliations:** Department of Finemechanics, Graduate School of Engineering, Tohoku University, Sendai 980-8579, Japan; ishikawa@bfsl.mech.tohoku.ac.jp

**Keywords:** spermatozoa, viscoelastic fluid, Stokes flow

## Abstract

It has been suggested that the swimming mechanism used by spermatozoa could be adopted for self-propelled micro-robots in small environments and potentially applied to biomedical engineering. Mammalian sperm cells must swim through a viscoelastic mucus layer to find the egg cell. Thus, understanding how sperm cells swim through viscoelastic liquids is significant not only for physiology, but also for the design of micro-robots. In this paper, we developed a numerical model of a sperm cell in a linear Maxwell fluid based on the boundary element slender-body theory coupling method. The viscoelastic properties were characterized by the Deborah number (*De*), and we found that, under the prescribed waveform, the swimming speed decayed with the Deborah number in the small-*De* regime (*De* < 1.0). The swimming efficiency was independent of the Deborah number, and the decrease in the swimming speed was not significantly affected by the wave pattern.

## 1. Introduction

The swimming of spermatozoa is responsible for the reproduction process, especially for finding egg cells. The human spermatozoon is about 50–100 μm [1], and has one long flagellum. By beating the flagellum, the sperm can propel into surrounding liquid, with an average swimming speed on the order of *O*(10^−5^) to *O*(10^−4^) m/s [1,2]. Recently, sperm-inspired micro-robots have been developed by researchers in the field of biomedical engineering and biomimetics [3,4]. Thus, understanding how sperm swim is important not only for physiology, but also for the fabrication of micro-robots.

Over small scales, the viscous effect can dominate when the inertial effects become sufficiently small. Thus, the locomotion of sperm cells is governed by micro-fluid dynamics. Several researchers have studied how sperm swim from the perspective of fluid mechanics. Smith et al. [5] numerically investigated sperm cells swimming near a planar wall. They showed that the cells accumulated at the wall and, due to the conical envelope of the flagellar trajectory, became aligned with it. The equilibrium height and stability were analyzed by Ishimoto and Gaffney [6]. Gadêlha et al. [7] investigated the fluid–structure interaction of flagellar motions and reported that nonlinear flagellar dynamics are essential for breaking the symmetry of the waveform. Kantsler et al. [8] experimentally observed the rheotaxis of sperm cells, which is self-organized upwards, swimming in the fluid flow. This mechanism can be explained in terms of fluid mechanics [9,10]. More detail on the fluid mechanics of sperm cells is provided in [1,11].

Mammalian sperm cells must swim through mucus layers in the genitals, such as the oviduct and the uterus. Mucus has viscoelastic properties [1]. Ishimoto et al. [12] investigated the locomotion of sperm cells through a viscoelastic fluid. They observed small yawing when the sperm were swimming through a viscoelastic medium, related to the greater density of the regularized point-force singularity along the flagellum. Wróbel et al. [13] developed a model of a flagellar swimmer moving through a viscoelastic network, on a length scale comparable to that of the swimmer. They concluded that stiffer networks enhance the velocity of the swimmer. Although a number of researchers have investigated sperm cells swimming through a viscoelastic liquid, many questions remain unanswered. In particular, the efficiency and locomotive power of sperm in a viscoelastic fluid have not been studied. As viscoelastic fluid can easily be found in the human body, knowledge about the efficiency of locomotion through such media should be improved to further our understanding of reproduction and our ability to fabricate micro-robots capable of operating in biological environments.

In this study, we carried out a numerical investigation of sperm cells swimming through viscoelastic fluid. The viscoelastic fluid was modeled by a linear Maxwell fluid. We evaluated the swimming speed and efficiency whilst varying the Deborah number. In Section 2, we explain the equations governing the motion and our numerical algorithm. Our results are presented in Section 3. Finally, we conclude the paper in Section 4.

## 2. Governing Equations and Numerical Methods

In this section, we explain the equations governing the motion of the sperm as well as our numerical methods. Details of the fluid mechanics of a Maxwell fluid in the low-Reynolds number regime can be found in [14,15], and we explain it briefly here.

### 2.1. Green’s Function for a Linear Maxwell Fluid

We assumed that a sperm cell was immersed in an unbounded linear Maxwell fluid. We estimated the Reynolds number of the sperm swimming through the fluid to be much smaller than unity [1]. Hence, we assumed that the fluid motion was a Stokes flow. The deviatoric stress tensor of the Maxwell fluid is defined as:(1)$$\left(1+\lambda \frac{\partial }{\partial t}\right)\tau =2\mu D,$$
where ***τ*** is the deviatoric stress of the fluid, *λ* is the relaxation time, and *μ* is the viscosity. ***D*** is the rate of deformation tensor, which is defined as:(2)$$D_{ij}=\frac{1}{2}\left(\frac{\partial v_{i}}{\partial x_{j}}+\frac{\partial v_{j}}{\partial x_{i}}\right),$$
where *v* is the velocity of the fluid. By introducing the invertible linear operator $L=\mu {\left(1+\lambda \partial /\partial t\right)}^{-1}$, the momentum equation of the Maxwell fluid with a point force ***F*** can be written as [15]:(3)$$-\frac{\partial p}{\partial x_{i}}+L\frac{\partial^{2}v_{i}}{\partial x_{k}\partial x_{k}}+\delta \left(x,t;x_{s},t_{s}\right)F_{i}=0,$$
where *p* is the pressure and *δ*(***x***,*t*) is the delta distribution function in space and time, so that the point force ***F*** is concentrated at a certain point ***x****_s_* and time *t_s_*. By using Green’s functions, the singularity solution of Equation (3) is given by [15]:(4)$$-\frac{\partial P_{j}}{\partial x_{i}}F_{j}+L\frac{\partial^{2}J_{ij}}{\partial x_{k}\partial x_{k}}F_{j}+8\pi \delta \left(x,t;x_{s},t_{s}\right)\delta_{ij}F_{j}=0,$$
where P(***x***,*t*; ***x****_s_*,*t_s_*) and ***J***(***x***,*t*; ***x****_s_*,*t_s_*) are the pressure kernel and the Stokeslet for the Maxwell fluid, respectively, which are singular at ***x****_s_* and *t_s_*:(5)$$P_{j}\left(x,t;x_{s},t_{s}\right)=P_{{}_{j}}^{0}\left(x;x_{s}\right)\delta \left(t-t_{s}\right)$$
and
(6)$$J_{ij}\left(x,t;x_{s},t_{s}\right)=L^{-1}J_{ij}^{0}\left(x;x_{s}\right)\delta \left(t-t_{s}\right),$$
where $P^{0}$ (***x***; ***x****_s_*) and ***J***^0^ (***x***; ***x****_s_*) are the Green’s functions of a Newtonian fluid:(7)$$P_{{}_{j}}^{0}\left(x;x_{s}\right)=2\frac{r_{j}}{r^{3}}$$
and
(8)$$J_{ij}^{0}\left(x;x_{s}\right)=\frac{\delta_{ij}}{r^{3}}+\frac{r_{i}r_{j}}{r^{3}},$$
where *r* = |***r***| and ***r*** = ***x*** – ***x****_s_*.

### 2.2. Boundary Integral Equation

To describe the fluid dynamics of a sperm cell in viscoelastic fluid, we now introduce the boundary integral equation. We were required to define an initial condition due to the presence of the time derivative in Equation (4). We assumed, as a starting condition, that the dynamics were trivial for *t* < 0 and that we had Stokes dynamics for *t* = 0 [15]. For *t* = *t_s_* (> 0), the flow field was calculated by integrating the singularity solution with respect to time and space:(9)$$v_{i}\left(x,t_{s}\right)=-\frac{1}{8\pi }\int_{0}^{t_{s}}\int_{S}L^{-1}J_{ij}^{0}\delta \left(t-t_{s}\right)q_{j}dSdt,$$
where ***q*** is the traction force per unit area, and *S* represents the surface of the cell.

In this study, the slender-body theory [16] was applied to the motion of the flagellum. The flagellum was modeled as a thin, curved rod with a length–radius ratio *a*/*L* = 3.57 × 10^−3^ [1], where *a* is the flagellar radius and *L* is the length. Then, the flow field can be described by the following boundary integral—the slender-body coupling formulation:(10)$$v_{i}\left(x,t_{s}\right)=-\frac{1}{8\pi }\int_{0}^{t_{s}}\left(\int_{head}L^{-1}J_{ij}^{0}\delta \left(t-t_{s}\right)q_{j}dS+\int_{flagellum}L^{-1}K_{ij}^{0}\delta \left(t-t_{s}\right)f_{j}dl\right)dt,$$
where ***f*** is the force per unit length of the flagellum, and ***K***^0^ is the slender-body kernel [16]. The delta distribution gives the limiting value of the time integral, and the boundary integral of the Maxwell fluid is reduced to the Newtonian kernel with the leading time derivative:(11)$$v_{i}\left(x,t_{s}\right)=-\frac{1}{8\pi }L^{-1}\left(\int_{head}J_{ij}^{0}q_{j}dS+\int_{flagellum}K_{ij}^{0}f_{j}dl\right).$$

### 2.3. Flagellar Waveform and Head Geometry

To accurately describe the flagellar waveform, we introduce an orthonormal body frame, *ξ_i_*, with the origin at the head–tail junction. We assumed a rigid connection between the head and the flagellum, and the minus *ξ*_1_-axis was set to coincide with the orientation vector. The waveform was specified a priori for a given transformation in terms of the following sinusoidal function [17]:(12)$$\begin{array}{l} \xi_{2}=A\left(\xi_{1}\right)\cos2\pi \left(\xi_{1}/L\eta \left(\xi_{1}\right)-t/T\right),\\ \xi_{3}=\alpha A\left(\xi_{1}\right)\sin2\pi \left(\xi_{1}/L\eta \left(\xi_{1}\right)-t/T\right), \end{array}$$
where *T* is the period of the flagellar waveform, *η* is the wavelength (which can vary with respect to *ξ*_1_), and *α* is the chirality parameter. These parameters were selected such that they coincided with experimental observations.

An experimental study of the waveform of the human sperm flagellum was reported in [18,19]. In low-viscosity liquid, high apparent curvatures appeared along with whole flagellum, and the head made repeated yawing motions. In high-viscosity liquid, on the other hand, a steeping wave appeared near the tip of the flagellum and the yawing motion of the head was less pronounced. We defined two different beating patterns to express these different wave modes, as shown in Figure 1, by varying the wavelength *η* [17]: *η* = 1 for Mode 1, and *η* = 0.9613 − 0.038 tan^−1^(5*ξ*_1_ − 4) for Mode 2. The amplitudes of the two waves were adjusted to match the swimming speed in a Newtonian fluid. If the chirality parameter, α, is also set to α = 0.25 [18], then the sperm makes rolling motions as it swims.

Both human and bull sperm are similar to asymmetric ellipsoids [1]. We mimicked the elliptical sperm head using the following mapping function [10]:(13)$$X_{1}^{sp}=X_{1},\;X_{2}^{sp}=\frac{c_{2}X_{2}}{c_{1}+c_{2}-X_{1}/\ell },\;X_{3}^{sp}=\frac{X_{3}}{c_{3}+X_{1}/\ell },$$
where *X^sp^* is a point on the head of the sperm, *X* is a point on the sphere with radius ℓ, and *c*_1_, *c*_2_, and *c*_3_ are non-dimensional shape parameters. The parameters were set to ℓ/L = 4.17 × 10^−2^, *c*_1_ = 3.0, *c*_2_ = 2.0, and *c*_3_ = 4.0 to ensure that the morphology accurately reflected a human sperm cell.

### 2.4. Numerical Procedure

To express the locomotion of the sperm cell, we assumed that the cell moves rigidly. One motion of the flagellum is defined by Equation (12). The velocity at a point *x_s_* on the cell can be decomposed as:(14)$$v\left(x_{s}\right)=V+\Omega \wedge \hat{r}+v^{fla}\left(x_{s},t\right),$$
where ***V*** is the translational velocity, **Ω** is the angular velocity, $\hat{r}=x_{s}-x_{g}$, and ***x**_g_* is the center-of-mass of the head. The velocity of the flagellum wave is ***v**^fla^* (***x**_s_*,*t*), a function of the point of interest and time. When ***x**_s_* is located on the cell head, then ***v**^fla^* is equal to zero. When ***x**_s_* is located on the flagellum, then ***v**^fla^* is determined by the time derivative of Equation (12).

We solved the resistance problem defined in Equations (11) and (14) with respect to the unknowns ***V***, **Ω** and the tractions ***q*** and ***f***. The head and flagellum were discretized into 1280 triangular meshes and 200 curved segments by spline interpolation. The surface integral of Equation (11) was solved using Gaussian numerical integration, while the finite time difference scheme was applied to the time derivative. We assumed the following force-free and torque-free conditions so that we could express the free-swimming sperm cell as:(15)$$\int qdS+\int fdl=0,\;\int q\wedge \hat{r}dS+\int f\wedge \hat{r}dl=0.$$The above equations were also discretized so that Gaussian numerical integration could be performed. Once ***V*** and **Ω** were determined, the point of interest was updated using the second-order Runge–Kutta method. For more detail about the numerical method, please refer to our previous study [10].

We then introduced an important parameter, the Deborah number, which represents the elasticity and viscosity ratio of the viscoelastic fluid, and is characterized by the relaxation time *λ*. In this study, the Deborah number is defined as *De* = *ωλ*, where *ω* is the angular frequency of the flagellar beat. In the small-Deborah number regime, *De* << 1, the mechanical properties of a Maxwell fluid asymptotically approach those of a Newtonian fluid. Due to the numerical instability, the maximum Deborah number is limited to be smaller than unity in this study. Although maximum *De* is limited, a weak viscoelastic liquid can be easily found in vitro situations [1,2,12]. For example, in an in vitro experiment of sperm swimming in 1% methylcellulose [12], the Deborah number was estimated to be *De* ≈ 0.2. We thus decided to vary *De* in the range 0 ≤ *De* ≤ 1.

## 3. Results and Discussion

### 3.1. Swimming Motion of a Sperm Cell in a Newtonian Fluid

Firstly, we investigated a sperm cell swimming through a Newtonian fluid. In the Stokes flow regime, the viscosity of the fluid simply functions as a multiplier for the force and traction. We can then take *μ* to be unity without loss of generality. Cell profiles from one period, with different flagellar beats, are shown in Figure 2a,b. In the case of Mode 1, the flagellar motion was similar to a traveling sinusoidal wave, whereas in Mode 2, the flagellar beat was more forceful at the tip. The amplitude of the beat in the middle part of the flagellum was comparable between the two modes. In both cases, the sperm cell exhibited a rolling motion before the trajectories began to follow a left-handed helicoid. The trajectories of the center-of-mass of the head in the (*x*,*y*)-plane are shown for 20 beats in Figure 2c. The initial swimming orientation was set to (1,0,0) for both beat modes. The head oscillated less in the direction perpendicular to the swimming motion. Although the direction in which the sperm swam was almost straight, small deviations in the *y*-direction were opposite in the two modes. This difference was caused by the initial condition. To facilitate quantitative discussion of the swimming speed, we define two types of average swimming speed. The swimming speed $\bar{U}$ was defined as the time average of the magnitude of the instantaneous translational velocity:(16)$$\bar{U}=\frac{1}{nT}\int_{t_{0}}^{t_{0}+nT}\left|v\right|dt,$$
where *n* represents the number of periods *T*. The latter swimming speed $\bar{V}$ was defined in terms of the displacement of the sperm cell over a sufficiently long time, shown with dotted arrows in Figure 2, that is,
(17)$$\bar{V}=\frac{\left|x\left(t_{0}+nT\right)-x\left(t_{0}\right)\right|}{nT}.$$

The swimming speeds $\bar{U}$ and $\bar{V}$ of the two beat modes are shown in Figure 2d. As the trajectories were almost straight, the values of $\bar{U}$ and $\bar{V}$ were similar between both beat modes, as shown in Figure 2d.

### 3.2. Swimming Motion of a Sperm Cell in a Maxwell Fluid

Next, we considered a sperm cell swimming through a linear Maxwell liquid. Since the impact of the initial conditions decays over the timescale of the relaxation time *λ*, we limit our discussion to the physical values observed after the first period *T*, *t* ≥ *T*, which was sufficiently large for the initial conditions to decay.

The temporal flow fields and pressure distributions at time *t* = *T* with *De* = 0.0 and *De* = 1.0 are shown in Figure 3. Note that the pressure field was determined by the kernel shown in Equation (5). As shown in Figure 3, high pressure was observed in the forepart of the flagellum/head movement, while the negative pressure appeared at the rear in all cases. In the case of wave mode 1 with *De* = 0.0 (Figure 3a), the pressure was enhanced within the region of *r*/*L* < 0.1 along with the whole flagellum, that is, the red region in Figure 3, where *r* is the distance from the nearest flagellum/head surface and *L* is the flagellar length. By increasing the Deborah number, the high-pressure region shrank to near the surface (*r*/*L* < 0.05, see Figure 3b). Accordingly, the flow generated by the sperm’s swimming diminished as the Deborah number increased. The same tendency was observed in wave Mode 2 (Figure 3c,d), although both flow and pressure fields were different between the two modes. In this study, the flagellum waveform was prescribed, that is, the beating speed was defined, and the traction was determined based on the time derivatives according to Equation (11). The traction force decreased with the flagellar velocity. As small traction forces cannot produce strong fluid flow, the flow was weak in the high-*De* regime.

The average swimming speeds $\bar{U}$ with different values of *De* are shown in Figure 4. The value was averaged in the range 10*T* ≤ *t* ≤ 20*T* and normalized by the swimming speed through a Newtonian fluid ${\bar{U}}_{0}$. In the small-Deborah number region *De* << 1, the swimming speed converged to ${\bar{U}}_{0}$. Interestingly, we obtained similar results for the two different swimming modes, with both values decaying with *De*. Given that the swimming modes differ, why do the swimming speeds vary in the same way? In the next section, we considered the power and efficiency so that we could identify the mechanism that caused the swimming speed to decay.

### 3.3. Power and Swimming Efficiency

To see the effect of the Deborah number in more detail, we investigated the power generated by the locomotion. The power *P* due to the cellular locomotion is:(18)P=∫v⋅qdS+∫v⋅fdl.

The time-averaged powers for the different wave modes are shown in Figure 5a. For all values of *De* tested, the power *P* of Mode 1 was always higher than that of Mode 2. However, the two curves matched when we normalized the value by the power when *De*
→ 0 (i.e., at *P*_0_), as shown in Figure 5b. We then investigated the efficiency of the locomotion. The efficiency was defined by the swimming speed per unit power, *μfL*^2^*U*/*P*, where *μ* is the viscosity, *f* is the beat frequency, and *L* is the flagellar length. The results are shown in Figure 6, where we can clearly see that the efficiency was almost independent of *De*, and the value became constant. When the efficiency was fixed, the tendency of the swimming speed to decay with *De* was less dependent on the wave pattern. Thus, the swimming speed varied in the same way in both wave modes.

## 4. Conclusions

In this study, we numerically investigated a sperm cell swimming in a linear Maxwell fluid in the small-*De* regime (*De* < 1.0). We found that, for the given waveform, the efficiency of the motion remained constant as the Deborah number varied. The fixed efficiency diminished the effect of the wave pattern on the decrease in swimming speed with *De*. These findings could be relevant to one-way fluid–structure interaction and may be helpful to researchers designing micro-swimmers in viscoelastic fluids with prescribed velocity conditions. For further study, to understand the physiology of sperm swimming, we must consider full fluid–structure interactions by developing a mechanical model of the inner structure of the flagellum.

## Figures and Tables

**Figure 1 micromachines-10-00078-f001:**
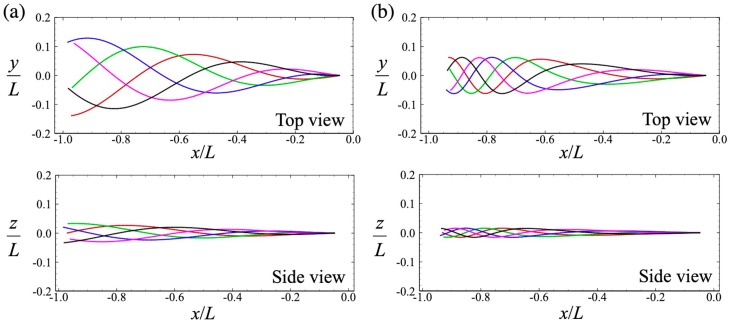
Flagellar beat with (**a**) Mode 1 and (**b**) Mode 2. Top panels are seen from the top, whereas bottom panels are seen from the side. Temporal flagellar profiles are shown by different colors: red for *t*/*T* = 0.0, green for *t*/*T* = 0.2, blue for *t*/*T* = 0.4, magenta for *t*/*T* = 0.6, and black for *t*/*T* = 0.8, where *T* is the beat period.

**Figure 2 micromachines-10-00078-f002:**
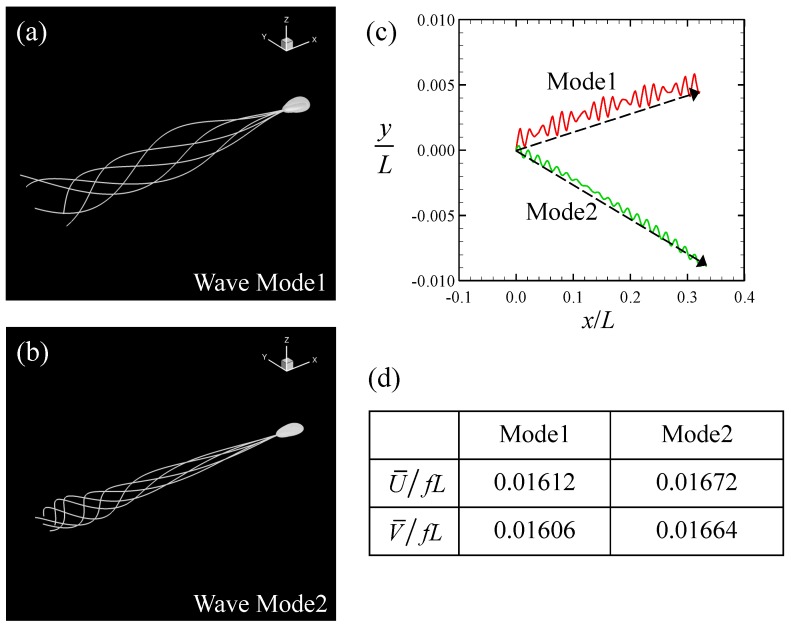
Swimming of the sperm model in a Newtonian fluid. (**a**) Superimposed image of the model during one period with Mode 1. (**b**) Superimposed image with Mode 2. (**c**) Trajectories of the center-of-mass of the head in the (*x*,*y*)-plane during 20 beats. The dotted arrow indicates the average in-plane displacement. (**d**) Time-averaged swimming speeds with two different definitions. The values are averaged across 20 beats and normalized by the beat frequency *f* (= 1/*T*, where *T* is the beat period) and the flagellar length *L*.

**Figure 3 micromachines-10-00078-f003:**
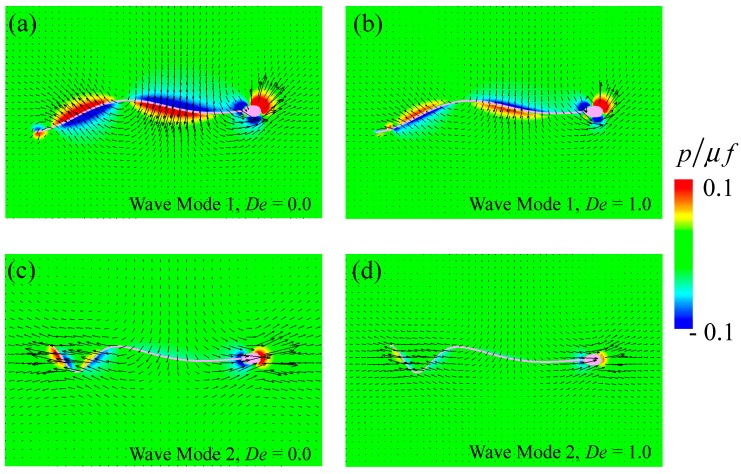
Temporal flow and pressure fields at *t* = *T* with *De* = 0.0 and 1.0; (**a**,**b**) show wave Mode 1 and (**c**,**d**) show wave Mode 2. The contours represent pressure, which is normalized by the viscosity *μ* and frequency *f*.

**Figure 4 micromachines-10-00078-f004:**
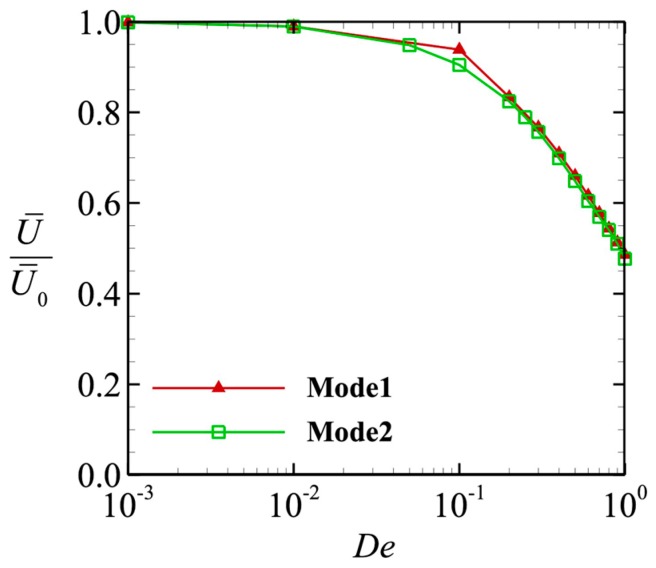
Time average swimming speed as a function of the Deborah number. The value was normalized by the swimming speed in a Newtonian fluid.

**Figure 5 micromachines-10-00078-f005:**
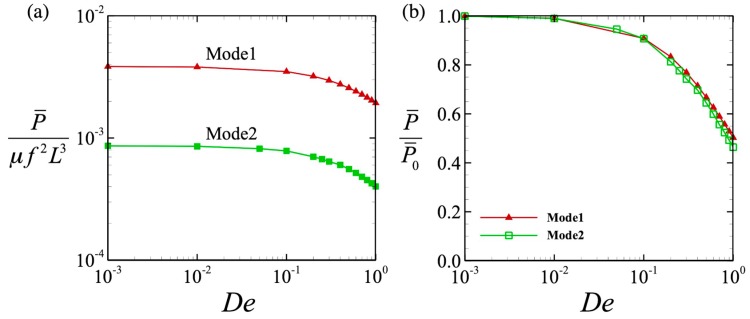
Power due to the locomotion: (**a**) The value was normalized by the viscosity *μ*, the beat frequency *f*, and the flagellar length *L*. (**b**) Power was normalized by *P*_0_ in a Newtonian fluid.

**Figure 6 micromachines-10-00078-f006:**
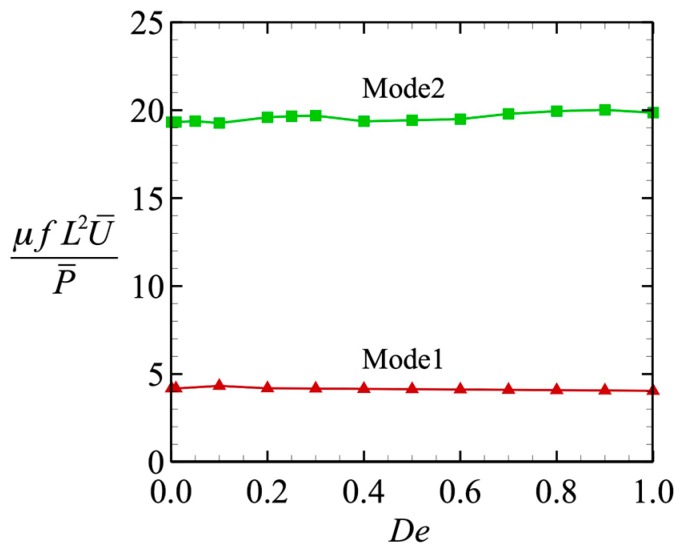
Swimming efficiency as a function of *De*.

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
