# Peer review of "Swimming of Spermatozoa in a Maxwell Fluid"

_micromachines, 2019, doi:10.3390/mi10020078_

Round 1

Reviewer 1 Report

The authors investigated flagella swimming in viscoelastic fluid, which is an interesting topic for active matter due to its potential applications in biology and guidance for the design of micromachines. The presentation is clear and easy to follow. However, there are some questions the authors need to clarify before I can recommend publication in Micromachines.

1.     The swimming speed  and are not good quantities to describe the two different modes of motion. One cannot see the major difference of the two modes of motion from these two quantities. Also do flagella have other Modes of motion?

2. The description of the influence of De number on the flow field is not clear (i.e., second paragraph in 3.2). The authors use contradicting words describing the far field and near cell flow field. Please clarify.

3. What is “T” in the first paragraph of 3.2. No definition for the first time it appears. There are also other symbols which are not defined the first time they show up. Please correct that.

4. In Figure 3, the authors show the flow field at different De number for the tow modes. However, the flow field has neither direction nor strength.

Author Response

We upload a pdf file which includes point-by-point response to the reviewe's comment.

Reviewer 2 Report

In the present work, the authors describe a numerical model of sperm cell motion in a non-Newtonian fluid. The originality and the potential impact of the research is high. Nonetheless, I have the following major comments before recommending for publication:

1.       It is confusing that the researchers develop their model for a viscoelastic (essentially non-Newtonian) fluid, as implied in the paper title, yet their main focus in the model was the low Deborah number, which exhibits Newtonian characteristics. 

2.       Because the major axis of the paper is to contribute to the understanding of andulatory sperm-like motion in the non-Newtonian fluids, why did the authors not try higher Deborah numbers, which is likely more representative fluid of the physiological mucous? The real contribution of the work to the literature needs to be better dissected.

Author Response

(The authors gave the same response as above.)

Reviewer 3 Report

Paper presents one of available methods of simulation of one way structure - fluid interaction.

Formally mathematical model and numerical solution are correct under assumed simplifications.

The main complain is related to the assumed model.

The head is rigidly connected with flagellum changing the space form in time in a predefined way.

The idea of the paper is to show the influence of swimming in non Newtonian fluid in comparison with Newtonian fluid.

In reality the shape of flagellum depends on the mass and stiffness of the flagellum fragments and reaction of fluid. Using two fluids, one Newtonian and second non-Newtonian the local fluid interaction with solid should be different as well the flagellum deformation. Simply - temporal flagellum shape should be different in both fluids.
Author assumes that is the same. Maybe the difference is not critical but exists.
If you notice that generated power is reduced two times between De=0 and De=1.0 it seems that interaction with real flagellum should changed it's shape (mode of motion) especialy at De =1.

Fig. 3 presents temporal flow field which means nothing. Is it temporal velocity field? Rather unclear visualization.

Fig. 2 c. It is not clear for me, why in Mode1, moving object is deviating left, and in Mode 2 is deviating right. If the sign in equation (12) defining ksi_3 will be changed, the direction of deviation will change?

In my opinion the main achievement presented in the paper is the presentation of relation between the modes of flagellum motion, generated power and efficiency and fluid properties represented by value of De. The only limitation of presented results is assumed one way structure - fluid interaction, which is in contradiction with reality.

I suggest to print a paper with minor modification clarifying presented above complains.

I suggest to notice existence of following papers not included in list of references.

Kenta Ishimoto, Hermes Gadelha, Eamonn A. Gaffney, David J. Smith, Jakson Kirkman-Brown, Human sperm swimming in a high viscosity mucus analogue, Journal of Theoretical Biology 446 (2018) 1-10

B.M. Friedrich, I.H. Riedel-Kruse, J.Howard, FF. Juelicher, 2010, High-precision tracking of sperm swimming fine structure provides strong test of resistive force theory, The Journal of Experimental Biology 213, 1226-1234.

Jens Elgeti, R.G. Winkler, Gerhard Gompper, Physics of Microswimmers - Single Particle Motion and Collective Behavior, Reports on Progress in Physics, April 2015.

Author Response

(The authors gave the same response as above.)

Round 2

Reviewer 1 Report

After all the changes made, I recommend the publication of the paper in Micromachines.

Reviewer 2 Report

The authors have clarified the points I raised. Therefore, I recommend publication as is.

Reviewer 3 Report

Explanation accepted.